# Ecological Momentary Assessment to Obtain Signal Processing Technology Preference in Cochlear Implant Users

**DOI:** 10.3390/jcm11102941

**Published:** 2022-05-23

**Authors:** Matthias Hey, Adam A. Hersbach, Thomas Hocke, Stefan J. Mauger, Britta Böhnke, Alexander Mewes

**Affiliations:** 1Audiology, ENT Clinic, UKSH, 24105 Kiel, Germany; britta.boehnke@uksh.de (B.B.); alexander.mewes@uksh.de (A.M.); 2Research and Development, Cochlear Limited, Melbourne, VIC 3000, Australia; ahersbach@cochlear.com; 3Research, Cochlear Deutschland, 30625 Hannover, Germany; thocke@cochlear.com; 4Research, Seer Medical, East Melbourne, VIC 3002, Australia; stefan.mauger@seermedical.com

**Keywords:** cochlear implant, signal processing, hearing in noise, EMA, ecological momentary assessment, acoustic environment, BEAM, ForwardFocus

## Abstract

Background: To assess the performance of cochlear implant users, speech comprehension benefits are generally measured in controlled sound room environments of the laboratory. For field-based assessment of preference, questionnaires are generally used. Since questionnaires are typically administered at the end of an experimental period, they can be inaccurate due to retrospective recall. An alternative known as ecological momentary assessment (EMA) has begun to be used for clinical research. The objective of this study was to determine the feasibility of using EMA to obtain in-the-moment responses from cochlear implant users describing their technology preference in specific acoustic listening situations. Methods: Over a two-week period, eleven adult cochlear implant users compared two listening programs containing different sound processing technologies during everyday take-home use. Their task was to compare and vote for their preferred program. Results: A total of 205 votes were collected from acoustic environments that were classified into six listening scenes. The analysis yielded different patterns of voting among the subjects. Two subjects had a consistent preference for one sound processing technology across all acoustic scenes, three subjects changed their preference based on the acoustic scene, and six subjects had no conclusive preference for either technology. Conclusion: Results show that EMA is suitable for quantifying real-world self-reported preference, showing inter-subject variability in different listening environments. However, there is uncertainty that patients will not provide sufficient spontaneous feedback. One improvement for future research is a participant forced prompt to improve response rates.

## 1. Introduction

Cochlear implantation is an established treatment option for patients with severe to profound, or moderate sloping to profound, bilateral sensorineural hearing loss [1,2]. To assess patients hearing ability and the success of cochlear implantation, speech perception is assessed through well-established tests performed in controlled conditions of the laboratory. Initially, speech perception was assessed with sentences in quiet [3,4], but assessment was complemented or replaced by more difficult word in quiet tests as cochlear implant (CI) patient performance increased [4,5]. Nowadays, monosyllabic or phoneme scores are an accepted measure used to identify and refer candidates for cochlear implantation [6,7] as well as for predicting and evaluating cochlear implant outcomes [8,9,10].

Speech perception in noise tests have also become a common outcome assessment, due to continued performance improvement in cochlear implant performance brought about by algorithms able to improve the signal-to-noise-ratio [11,12,13,14,15,16]. These tests also support further development and evaluation of new algorithms involved in cochlear implant processing, the access of CI recipients to sound processor upgrades through demonstrated performance improvements [14,17,18], and the individualization of settings in sound processors [12]. The assessment of the potential benefit of recent algorithms such as ForwardFocus (Cochlear Limited, Sydney, Australia) [18] expand the boundaries of current clinical audiometry practice.

Algorithms like ForwardFocus are designed to improve speech perception in complex real-world listening environments, where the target speech is in front of the listener and multiple and dynamic competing signals are towards the side and/or the rear [11,17,18,19,20]. These are challenging environments to simulate in a test booth, as they require significantly more dedicated hardware and software than commonly available in clinical audiometry practice. Questionnaires can assess the therapeutic effect through preoperative and postoperative comparison for a CI treatment or processor upgrade and can provide suitably complex listening environments for the evaluation of sound processor programs, which could include algorithms such as ForwardFocus [21]. However, data from questionnaires rely on retrospective recall of events and experiences and therefore reflect cumulative effects, are possibly biased by the interlocked effects of long-term memory and inference [22], and can therefore be inaccurate. Questionnaires also do not capture the variation of the sound environment across the day, a particular disadvantage in assessing algorithms designed for particular acoustic situations.

Clinical research in a variety of fields [23], and more recently in hearing research [24,25], has begun using a methodology called Ecological Momentary Assessment (EMA) to collect real-time situational responses from patients [26]. This method has the advantage of being conducted in real time in complex real-world situations, mitigating the limitations of common hearing research clinical outcome assessments. While EMA has been used in studies with hearing aid users [24,25,27], this method has not been widely used in studies with cochlear implant patients.

Most signal processing algorithms and fitting strategies in CI users are investigated in the lab and averaged over a group of patients. They do not take into account the individual needs and the time-dependent character of judging a given hearing program [28]. This evokes the need to validate these findings in real life. EMA methods have several advantages, for example, improved ecological validity due to data assessment in the real world; accounting for variations over time; being less vulnerable to recall bias [28]. Nevertheless, it has to be noted that this method is demanding and time consuming for subjects. Consequently, results may have variable reliability, as feedback is given without the presence of an investigator [28]. On the other hand, EMA methods allow the collection of time-dependent data, providing more detailed insights into the acoustic reality of CI patients in contrast to the questionnaire-based assessment when investigating in the clinic.

The audiometric clinical routine shows limitations in transferring the acoustic reality into an audiometric booth [27,29]. Additionally, it was shown that signal processing in sound processors should be individualized [12]. However, so far there is no method and no gold standard known to provide further detailed insights into patients views without extensive audiometric testing. To summarize, the evaluation of the individual benefit of signal processing algorithms expands the boundaries of current clinical audiometry practice [30].

The goal of this study was to investigate the feasibility of EMA in a CI population. The ability to capture specific data on the acoustic environment as well as patient-specific preference data on sound processing algorithms should be investigated. The individual preference of the new ForwardFocus algorithm [21], known to provide benefits in complex dynamic noise environments found in the real-world, is compared to the well-established Beamformer.

## 2. Materials and Methods

### 2.1. Research Subjects

This investigation included eleven (five unilateral and six bilateral) CI subjects. The patients were recruited from the clinic’s patient pool. The investigation was approved by the local ethics committee (D 467/16), and all procedures were in accordance with the ethical standards of the institutional and national research committee and with the 1964 Helsinki declaration and its later amendments or comparable ethical standards.

CI subjects were recruited who were at least 18 years of age, with post-lingual onset of deafness and implantation with a Nucleus CI24RE or CI500 series cochlear implant (Cochlear Limited, Sydney, Australia), and who were current users of a CP900 series sound processor (Nucleus 6^®^). All subjects had at least six months’ experience with their CI system. Bilateral implantation was not an exclusion criterion. Demographic information of these patients is provided in Table 1. This study cohort contained a subset of 20 subjects reported in Hey et al., 2019, who also took part in this additional EMA investigation. The signal processing algorithm ForwardFocus was evaluated in the laboratory in a range of noise types (stationary and fluctuating) as well as different spatial conditions (signal and noise from front; signal from front and noise from the posterior hemisphere) [21]. Reference for further comparison was the known BEAM algorithm [15,17]. It was shown that ForwardFocus was able to significantly improve speech comprehension in a wide range of acoustic scenes constructed in the laboratory.

### 2.2. Programming the Sound Processor Settings

During an initial session, participants were provided with two programs of the sound processor. The first program (subsequently named as “BEAM”) consisted of default Nucleus 6 SmartSound^®^ iQ technologies (ADRO, SNR-NR and ASC), with the addition of BEAM (adaptive directional microphone) [12,31]. The second program (“FF”) contained the same Nucleus 6 SmartSound iQ technologies, with the addition of the ForwardFocus technology [21] implemented for research. All other fitting parameters were the same for both programs. The patients’ MAPs were not changed for the study. Programs were randomized between the two program slots, and subjects were blind to the program slot allocation. To change programs and capture EMA data, a CI remote control (Nucleus^®^ CR230; Figure 1) was provided to each patient for the take-home period. Programs were simply labelled “1” and “2” in order of the program slots used.

### 2.3. EMA Data Capture and Analysis

The CR230 remote-control device allowed subjects control over sound processor volume and sensitivity, as is typical for daily use (Figure 1). It also displayed the current listening environment class [32]. A large side button (conventionally used to enable the telecoil feature) was repurposed and used as a vote button. The data logging capability of the CR230 allowed the listening environment (Quiet, Speech, Speech in Noise, Noise, Wind, and Music) and listening program to be recorded as the user pressed the vote button. These features provided a suitable platform to capture EMA data. In this study, we investigated a sound processing program preference through subject voting between a BEAM program and a ForwardFocus program in real-world environments. For analysis, the listening environments relevant for communication were used, which excluded the Wind and Music classes.

Subjects were provided with two programs and a sound processor remote control for a two-week period. During this period, they were asked to change between programs during each day to experience both programs. Subjects were also instructed to complete at least one vote (data capture) each day in a range of their different listening environments across the two-week period. To vote, subjects were instructed to change between programs during normal use of the device, and after several changes back and forth, to vote for their preferred program by pressing the side button on the remote control.

Data capture of the patient’s instantaneous listening environment was possible due to the SCAN scene classification algorithm available on the CP900 sound processor [32]. At each time instance, the environment is classified into one of six sound classes: Quiet, Speech, Speech in Noise, Noise, Music, and Wind. This algorithm is based on extracting acoustic features such as sound level, modulation, and frequency spectrum from the microphone signal, followed by a decision tree to determine the sound class [32,33]. A data-driven machine learning approach was used to train the decision tree using sound recordings labelled by humans with the appropriate sound class. In contrast to the commercially available CP900, during this study the classification system did not make any automatic changes to the sound processing or program selection but was only responsible for determining the sound class for the purpose of data logging.

At the end of the two-week period, data logs containing the vote events, scene classification data, and program selection were downloaded. Analysis was first performed to exclude accidental voting and exclude votes that did not show temporal coincidence with previous changes between both programs. In order to determine the sound class associated with each vote, the detected sound class was analyzed over the 10 s preceding the vote event. It was assumed that the evaluation of programs would likely have occurred over a period of time, possibly under different scene classifications. In cases where the sound class was variable, the vote was assigned according to the dominant sound class over the 10 s preceding the vote event, and in the case of an equal distribution, to the most recently detected sound class. The preferred listening program was determined from the listening program that was selected at the time the vote button was pressed.

For each subject, raw vote data were aggregated separately for each acoustic scene and represented in a program verse scene matrix, where each element represented the number of votes for each program.

Statistical analysis was performed in R statistics package version 4.1.1. Program preference (vote) was modelled as a binomial dependent variable using repeated measures logistic regression by fitting generalized linear models (glms) with the logistic link function.

## 3. Results

### EMA Results

A total of 205 valid votes were cast in total across the study group over the two-week period. The median number of votes cast by each subject was 15 and ranged from a minimum of seven to a maximum of 50. Six subjects voted at least once per day on average over the 2-week period, while five subjects voted less often. Votes were spread across the different acoustic scenes, the distribution of which is provided in Figure 2 for the entire subject pool. The scene with the fewest votes cast was Speech with 19 votes, while the other three classes had an approximately equal number of votes, with 55, 54 and 45 votes cast in the Quiet, Speech in Noise, and Noise class respectively. The median number of votes cast per subject in each scene was 6, 1, 4 and 4 for the Quiet, Speech, Speech in Noise, and Noise classes, respectively.

The number of votes cast by each individual subject is presented in Figure 3 using bubble plots. The location of the bubble on the x-axis indicates the program preference, the size of the bubble indicates the number of votes that contributed to that data point and the color indicates the sound class to which the votes were allocated.

Overall preference was analyzed by aggregating data across all scenes. A glm was fitted with program preference as the dependent variable and subject as the independent variable. The resulting chi-squared analysis of variance on the glm showed the effect of subject was highly significant (*p* < 0.001). P-values indicating the significance of preference for each subject are presented in Table 2. Two subjects had a significant preference for FF labelled as category A: subject #7 (*p* = 0.004) and #15 (*p* < 0.001). The remaining nine subjects showed no significant overall preference for either program.

To analyze the effect that sound class had on preference, a mixed-effects glm was fitted. The dependent variable was program preference and independent variable was sound class, while subject was considered as a random effect in the model.

The effect of SoundClass (fixed effect) was tested by comparing the mixed-effects glm to a model that excluded the SoundClass (fixed effect) and only included the random effect (subject). The resulting chi-squared analysis of variance showed SoundClass fixed effect was not significant (*p* = 0.191), indicating a lack of association between program preference and sound class for the group of subjects as a whole.

The effect of subject (random effect) was tested by comparing the mixed-effect glm to a model that excluded the random effect (Subject) and only included the fixed effect (SoundClass). The resulting chi-squared analysis of variance confirmed that Subject (random effect) was highly significant (*p* < 0.001), indicating that individual subjects voting preferences varied amongst the group. For each individual subject, the logistic regression of preference with SoundClass was fitted, and the resulting p-values are shown in Table 1. Three subjects showed logistic regression that was significant, indicating that those subjects voting preference was dependent on the SoundClass (#1 *p* = 0.004, #13 *p* = 0.011, #14 *p* = 0.017), labelled as category B.

Six subjects (#4, #6, #9, #12, #14, #17) were labelled as category C, for which no conclusive preference could be determined. Four of those subjects voted less than once per day on average over the 2-week period.

## 4. Discussion

The automatic scene classifier in the cochlear implant sound processor offers the ability to characterize the surrounding with respect to its acoustics characteristics, such as speech and noise. It is known that signal pre-processing in cochlear implant systems should be chosen depending on the acoustic environment to improve speech comprehension [32]. Such conclusions were derived from in-lab investigations. So far, there is limited knowledge available on the patient’s everyday real-world program preference using such algorithms.

The technical realization of the remote control of modern cochlear implant systems, as used as a scene-dependent voting tool in this study, can be used for the EMA in a cochlear implant population. The integration of the assessment tool into the patient’s sound processor proved to be useful. The resulting link of the patients’ input to the captured acoustic scene class potentially allows for the investigation of patients’ individual preferences with respect to program settings and/or specific algorithms in different acoustic environments. Additionally, in cases of data mismatch to clinical expectations or ongoing inactivity, this method may provide new insight into individual preferences. This pilot study showed a significant difference in voting patterns across the group of subjects: for instance, two patients (#15, #7) had an overall preference for ForwardFocus that persisted regardless of the acoustic scene, three subjects (#1, #13, #14) had a scene specific algorithm preference, and the remaining six subjects (#4, #6, #9, #10, #12, #17) had no conclusive preference.

Our methodology complements the use of EMA in hearing science to date, where studies have prompted surveys where participants assess their acoustic environment and rate their hearing experience [24,25,27]. These methodologies provide in-the-moment responses to complex real-world situations, which is a significant improvement to surveys confounded by retrospective recall. In addition, our approach enables in-the-moment rating of signal pre-processing technology for real-world environments.

A significant advantage of this EMA methodology is the objective acoustic scene classification. By using the available scene classification of the sound processor [31], an accurate environmental measure is captured without further patient interaction. This ability is expected to be useful for sound processing algorithms that are designed and expected to provide benefit in specific noise environments. Research algorithms are being developed for specific noise scenes, such as constant noise [34] or babble noise [35]. To complement the in-booth speech understanding results, this method could provide real-world preference results for each of the available scene classes.

This study also aimed to investigate the feasibility of EMA to provide data on sound processing algorithms. Two sound processing algorithms were compared, with one being the adaptive directional microphone BEAM, and the other being ForwardFocus, known to provide significant speech understanding over BEAM in dynamic noise environments. Although these technologies were chosen because they were expected to provide differences, particularly in noisy listening environments, no clear general or scene-specific difference was determined from the group. This is not unexpected, due to the patient numbers and the numbers of votes collected. What was found was some evidence of individualized general or scene-specific voting patterns. In future, such EMA studies should therefore consider the proportion of votes expected in each scene. For instance, in this study, patients were far less likely to vote in Wind, Music, or Speech in Noise scenes spontaneously. For a prompted methodology, the proportion of time, on average, in each scene would be important to consider and could be found from the sound processor data logs [36,37]. These insights will provide at least a basis to determine study design to power and capture data for direct scene-specific algorithm comparisons.

Compared to BEAM, the ForwardFocus algorithm shows its advantages in speech understanding, especially in fluctuating noise [21]. Several consequences can result from this. The acoustic scenes of Noise and Speech in Noise are not only characterized with respect to the signal-to-noise ratio, but a characterization of the temporal properties of the noise is additionally performed: stationary or fluctuating. This can be the basis to introduce ForwardFocus as an algorithm that is activated in specific listening scenes, such as determined by the automatic classification algorithm SCAN [18,32].

### Study Limits and Future Improvements

Subjects were asked to vote at least one time per day during the two-week take-home period. A total of 205 votes were recorded from the 11 patients, resulting in over one vote per subject per day, on average. All patients were able to use the remote control for data capture, successfully demonstrating the feasibility of EMA within a CI population. However, our self-initiated data capture method had some limitations. Six of the eleven patients provided at least one vote per day, while the remaining five subjects voted less often (averaged over the two-week period). Due to the low number of votes, four of those subjects had preference outcomes that were non-conclusive. In future such EMA studies, it would be beneficial to include a forced patient prompt, to request patients to conduct a comparison and vote. Additionally, an incentive, e.g., progress bars and gamification, for the study participant may help to minimize missing votes. Furthermore, the recipients should have the chance to withdraw an accidental vote. To support this, a possible review by the participants themselves of all votes might be worth considering for future studies. To summarize, most of the above-mentioned deficits cannot be addressed within a feasibility study.

The problem of obtaining an adequate number of responses is also described by Wu et al. [38]. The prompting frequency, take-home duration, number of acoustic scenes captured, and number of subjects need to be optimized to achieve sufficient data for subsequent statistical analysis. Nevertheless, a sufficient number of valid votes were collected in this study, showing the feasibility of EMA as well as this methodology in CI recipients.

The current fitting philosophy is to provide beneficial sound processing algorithms for each listening scene, which has been shown to provide benefits to a group, on average [31,39]. However, it may be that individual algorithm selection could provide further individual benefit [12]. These varying individual performance benefits or preferences can provide input for a machine learning approach [40] to select individualized sound processing algorithm options for specific listening scenes.

A central question for clinical application remains: is the individual scene-specific preference aligned with the benefits shown in speech-audiometry tests. The EMA methodology is expected to elucidate such real-world individualized patient algorithm preferences and may help to fit speech processing algorithms better to the individuals’ needs.

## 5. Conclusions

This study found that program preference varied significantly among subjects. Some demonstrated an overall program preference, others demonstrated scene-specific preferences, and others demonstrated no conclusive preference. The data collection tool was integrated into the patient’s cochlear implant system and was suitable for real-world self-assessment. To improve data collection, future research should encourage participants to realize a higher response rate.

## Figures and Tables

**Figure 1 jcm-11-02941-f001:**
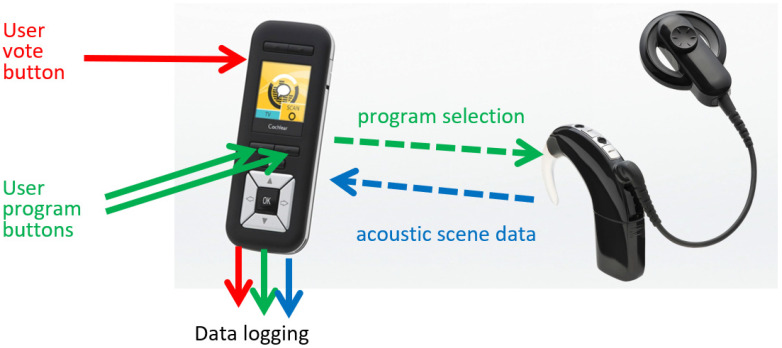
The CP900 sound processor and CR230 remote control used to capture EMA data.

**Figure 2 jcm-11-02941-f002:**
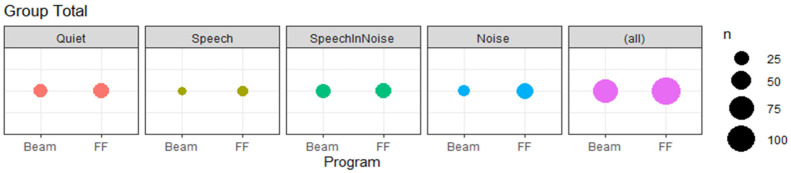
Program preference accumulated across entire subject group separated by sound class. Size of data point indicates number of votes. “BEAM” specifies the program consisting of the algorithms ADRO, SNR-NR, ASC and BEAM. “FF” indicates the second program containing the ForwardFocus microphone technology.

**Figure 3 jcm-11-02941-f003:**
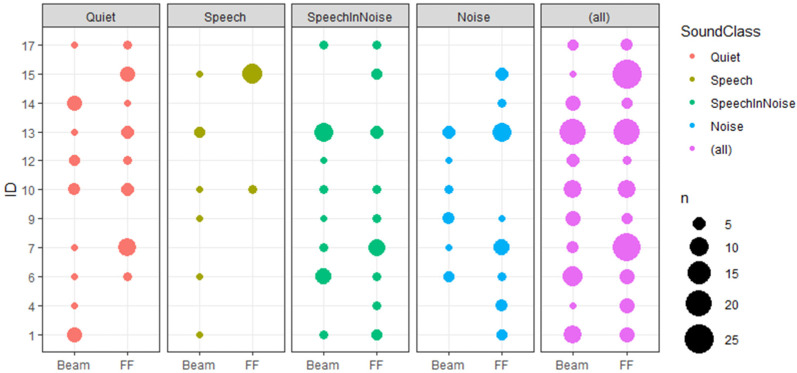
Individual program preference separated by sound class. Size of data point indicates number of votes.

**Table 1 jcm-11-02941-t001:** Biographical data of recipients.

Patient ID	Age (Years)	Usage of CI (Years)	Side	Gender	Rate (pps)	Maxima
#1	75.7	1.5	r	m	1200	12
#1	75.7	1.0	l	m	1200	12
#4	73.7	10.7	r	m	1200	8
#6	43.3	8.2	r	f	1200	12
#6	43.3	2.1	l	f	1200	12
#7	56.0	7.3	r	f	1200	12
#7	56.0	8.6	l	f	1200	12
#9	47.4	3.4	r	m	1200	12
#9	47.4	2.5	l	m	1200	12
#10	64.9	1.5	r	f	1200	12
#12	61.1	6.1	r	f	500	8
#12	61.1	8.7	l	f	500	10
#13	56.0	3.0	r	f	900	8
#14	65.0	10.9	r	f	500	12
#14	65.0	9.1	l	f	500	12
#15	73.4	2.6	l	m	900	10
#17	55.8	9.5	r	m	1200	12

**Table 2 jcm-11-02941-t002:** Summary of statistical analysis, indicating those subjects with an overall preference (A), those subjects whose preference varied with SoundClass (B), and those subjects where no preference could be determined (C). # xx – patient ID; * means significant test result.

Subject	Total Votes Cast	Logistic Regression of Preference with Subject (*p*-Value)	Logistic Regression of Preference with SoundClass (*p*-Value)	Category	Comments
*#13*	40	0.509	0.011 *	B	Preference varied with SoundClass
*#7*	28	0.004 *	0.807	A	Overall preference for FF
*#15*	27	<0.001 *	0.682	A	Overall preference for FF
*#6*	18	0.692	0.419	C	No conclusive preference
*#10*	18	0.566	0.358	C	No conclusive preference
*#1*	15	0.442	0.004 *	B	Preference varied with SoundClass
*#9*	9	0.744	0.268	C	No conclusive preference
*#14*	9	0.744	0.017 *	B	Preference varied with SoundClass
*#4*	7	0.068	0.057	C	No conclusive preference
*#12*	7	0.605	0.439	C	No conclusive preference
*#17*	7	0.455	0.658	C	No conclusive preference

## Data Availability

Not applicable.

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
