# Peer review of "Ecological Momentary Assessment to Obtain Signal Processing Technology Preference in Cochlear Implant Users"

_jcm, 2022, doi:10.3390/jcm11102941_

Round 1

Reviewer 1 Report

The authors provide a clear and well-written manuscript evaluating an Ecological Momentary Assessment (EMA) procedure for patients with cochlear implants.

The originality of the study is based in particular on the possibility of automatically collecting and classifying the patient's sound environment at the time of the vote, which provides relevant information compared to conventional evaluations in sound rooms, which are sometimes unrepresentative of everyday listening situations.

Nevertheless, several methodological aspects seem questionable.

(Lines 84-89) The two goals defined (primary and secondary) seem in fact very close even if their wording is slightly different. In both cases, the aim is to assess the “feasibility” of this type of EMA approach, a notion that is difficult to define. Is it a purely technical validation of the procedure, or an assessment of the reliability of the measures and their ability to report on patient performance? In the latter case, what is then the element of comparison or the "Gold Standard" used to assess the relevance of the data collected by EMA?

(Results/ Table 1). It appears that 5/11 subjects did not follow the instruction to make at least one vote per day. This makes their results difficult to interpret, and their data should perhaps have been excluded because the methodology presented was not followed. This lack of data may partly explain the great variability of the results and the fact that no preference (or category C) was found in 6/11 subjects, 4 of whom had not performed the requested minimum of 14 votes.

(Lines 142-147) The method and acoustic criteria to classify the sound environment into six categories are not explained. I think that even a brief description of the characteristics taken into account for this classification would be interesting for the reader.

The study by Wu et al. (Quoted line 284) had also highlighted the problem of reliability of this type of EMA approach in which we do not control the effective participation nor the context in which the subject votes. The problem of “accidental votes” (Line 153), the number of which should be mentioned, illustrates this difficulty. The study presented here confirms this lack of reliability and the problem of the missing data for the majority of patients. As a result, the general conclusion of the abstract (lines 26-29) should be, in my opinion, a little more nuanced on the relevance of this method given the limits of the results presented here.

References: citation 21 (line 374) is identical to citation 29 (line 392).

8/39 references with Hocke T (third author), including one duplicated

Author Response

The authors provide a clear and well-written manuscript evaluating an Ecological Momentary Assessment (EMA) procedure for patients with cochlear implants.

The originality of the study is based in particular on the possibility of automatically collecting and classifying the patient's sound environment at the time of the vote, which provides relevant information compared to conventional evaluations in sound rooms, which are sometimes unrepresentative of everyday listening situations.

Nevertheless, several methodological aspects seem questionable.

(Lines 84-89) The two goals defined (primary and secondary) seem in fact very close even if their wording is slightly different. In both cases, the aim is to assess the “feasibility” of this type of EMA approach, a notion that is difficult to define. Is it a purely technical validation of the procedure, or an assessment of the reliability of the measures and their ability to report on patient performance? In the latter case, what is then the element of comparison or the "Gold Standard" used to assess the relevance of the data collected by EMA?

  • We added a motivation for using this new method in comparison to standard clinical routine. It is difficult to compare it against a “Gold Standard” as there is no one. We tried to highlight this approach in relation to known literature.

“Audiometric clinical routine shows limitations to transfer the acoustic reality into an audiometric booth [27,29]. Additionally, it was shown that signal processing in sound processors should be individualized [12]. However, so far there is no method and no gold standard known to provide further detailed insights into patients view without extensive audiometric testing. To summarize, the evaluation of the individual benefit of signal pro-cessing algorithms expands the boundaries of current clinical audiometry practice [30].”

  • We agreed with you concerning the two goals. This might be misleading. We adapted the paragraph accordingly:

“The goal of this study was to investigate the feasibility of EMA in a CI population. The ability to capture specific data about the acoustic environment as well as patient specific preference data on sound processing algorithms should be investigated. The individual preference of the new ForwardFocus algorithm [21] known to provide benefit in complex dynamic noise environments found in the real-world will be compared to the well established Beamformer”

(Results/ Table 1). It appears that 5/11 subjects did not follow the instruction to make at least one vote per day. This makes their results difficult to interpret, and their data should perhaps have been excluded because the methodology presented was not followed. This lack of data may partly explain the great variability of the results and the fact that no preference (or category C) was found in 6/11 subjects, 4 of whom had not performed the requested minimum of 14 votes.

  • We agree that there is an association between small number of votes and category C (non-conclusive preference). While there is the possibility to completely remove them from the analysis, I would prefer to keep them in because it provides a statistical basis for the non-conclusive preference. Additionally, we did not state such removal prior to the data collection, and I can imagine a review opinion that would argue they should not be excluded, especially if an overall preference for FF was found as a result. The instruction to vote at least once per day was provided as general advice to subjects rather than a strict requirement. Therefore, it is my opinion that the statistical analysis of all subjects provides a complete picture of the data that was collected during the study. I have highlighted in the results and discussion the association between low number of votes and inconclusive preference outcome, and use it as motivation to suggest improved data collection in future.
  • Additionally a new order in table was introduced. Table was ordered in terms of total number of votes, to highlight the fact the non-conclusive preference was often associated with low number of votes.
  • Some sentences were added in results section.
  • Additional information was introduced in Study limits and future improvements:

“Subjects were asked to vote at least one time per day during the two-week take-home period.  A total of 205 votes were recorded from the 11 patients resulting in over one vote per subject per day on average.  All patients were able to use the remote control for data capture, successfully demonstrating the feasibility of EMA within a CI population. How-ever, our self-initiated data capture method had some limitations. Six of the 11 patients provided at least one vote per day, while the remaining five subjects voted less often (av-eraged over the two-week period). Due to the low number of votes, four of those subjects had preference outcomes that were non-conclusive. In future such EMA studies it would be beneficial to include a forced patient prompt to request patients to conduct a compari-son and vote. Additionally, an incentive, e.g. progress bars and gamification, for the study participant may help to minimize missing votes. Furthermore, the recipients should get the chance to withdraw an accidental vote. To support this, a possible review by the par-ticipants itself of all votes might be worth to con-sider for future studies. To summarize, most of the above mentioned deficits cannot be addressed within a feasibility study.”

(Lines 142-147) The method and acoustic criteria to classify the sound environment into six categories are not explained. I think that even a brief description of the characteristics taken into account for this classification would be interesting for the reader.

  • Thanks for this hint. We added some information and introduced new literature. Now it reads as follows:
    “This algorithm is based on extracting acoustic features such as sound level, modulation, and frequency spectrum from the microphone signal followed by a decision tree to deter-mine the sound class [32,33]. A data driven machine learning approach was used to train the decision tree using sound recordings labelled by humans with the appropriate sound class.”

The study by Wu et al. (Quoted line 284) had also highlighted the problem of reliability of this type of EMA approach in which we do not control the effective participation nor the context in which the subject votes. The problem of “accidental votes” (Line 153), the number of which should be mentioned, illustrates this difficulty.

  • We agree with you that accidental votes are a problem for this EMA method. We added some remarks in the discussion:
    Furthermore, the recipients should get the chance to withdraw an accidental vote. To support this, a possible review by the participants itself of all votes might be worth to con-sider for future studies.

The study presented here confirms this lack of reliability and the problem of the missing data for the majority of patients. As a result, the general conclusion of the abstract (lines 26-29) should be, in my opinion, a little more nuanced on the relevance of this method given the limits of the results presented here.

  • We modified the paragraph “Study limits and future improvements” with respect to Your comments. We put more effort in highlighting the shortcomings of our studies in order to provide tips and possible improvements for future studies.
    Now the abstract reads as follows:

“Results show that EMA is suitable for quantifying real-world self-reported preference, showing inter-subject variability in different listening environments.  However, there is uncertainty that patients will not provide sufficient spontaneous feedback.  One improvement for future research is a participant forced prompt to improve response rates.”

References: citation 21 (line 374) is identical to citation 29 (line 392).

  • Thanks for this hint. We corrected it.

8/39 references with Hocke T (third author), including one duplicated

  • Thanks for this hint. We corrected references.
  • We want to thank you for your effort.

Reviewer 2 Report

The presented paper is researching very important topic in audiology today – rehabilitation after cochlear implantation.

Clinical research in a variety of fields, and more recently in hearing research, have begun using a methodology called Ecological Momentary Assessment (EMA) to collect real-time situational responses from patients. This method has the advantage of being conducted in real-time in complex real-world situations, mitigating limitations of common hearing research clinical outcome assessments. While EMA has been used already in studies with hearing aid users, this method has not been widely used in studies with cochlear implant patients.

Speech perception in noise has also became a common outcome assessment in cochlear implant users, due to continued performance improvement in cochlear implant performance brought by algorithms which are able to improve signal-to-noise-ratio. These tests also support further development and evaluation of new algorithms involved in cochlear implant processing, the access of CI recipients to sound processors upgrades through demonstrated performance improvements, and the individualization of settings in sound processors.

The objective of this study was to determine the feasibility of using EMA to obtain in-the-moment responses from cochlear implant users describing their technology preference in specific acoustic listening situations. The primary goal of this study was to investigate the feasibility of EMA in a CI population, investigating its general use as well as its ability to capture specific data about the acoustic environment. The secondary goal was to investigate the feasibility of EMA to provide preference data on sound processing algorithms including the comparison to the new ForwardFocus algorithm known to provide benefit in complex dynamic noise environments found in the real-world.

The methodology used by authors complemented the previous use of EMA in hearing science to date, where studies prompted surveys where participants assess their acoustic environment and rated their hearing experience. methodologies used by authors provide in-the-moment responses to complex real-world situations, which is a significant improvement to surveys confounded by retrospective recall. In addition, authors approach enables in-the-moment rating of signal pre-processing technology for real-world environments.

The results of this study have shown that program preference varied significantly among subjects. Also, it was noticed that to improve data collection, future research should encourage participants to realize a higher response rate. It was concluded that EMA is suitable for quantifying real-world self-reported preference, showing inter-subject variability in different listening environments.

The presented paper opens up new research capabilities in patients with cochlear implants.

Author Response

The presented paper is researching very important topic in audiology today – rehabilitation after cochlear implantation.

Clinical research in a variety of fields, and more recently in hearing research, have begun using a methodology called Ecological Momentary Assessment (EMA) to collect real-time situational responses from patients. This method has the advantage of being conducted in real-time in complex real-world situations, mitigating limitations of common hearing research clinical outcome assessments. While EMA has been used already in studies with hearing aid users, this method has not been widely used in studies with cochlear implant patients.

Speech perception in noise has also became a common outcome assessment in cochlear implant users, due to continued performance improvement in cochlear implant performance brought by algorithms which are able to improve signal-to-noise-ratio. These tests also support further development and evaluation of new algorithms involved in cochlear implant processing, the access of CI recipients to sound processors upgrades through demonstrated performance improvements, and the individualization of settings in sound processors.

The objective of this study was to determine the feasibility of using EMA to obtain in-the-moment responses from cochlear implant users describing their technology preference in specific acoustic listening situations. The primary goal of this study was to investigate the feasibility of EMA in a CI population, investigating its general use as well as its ability to capture specific data about the acoustic environment. The secondary goal was to investigate the feasibility of EMA to provide preference data on sound processing algorithms including the comparison to the new ForwardFocus algorithm known to provide benefit in complex dynamic noise environments found in the real-world.

The methodology used by authors complemented the previous use of EMA in hearing science to date, where studies prompted surveys where participants assess their acoustic environment and rated their hearing experience. methodologies used by authors provide in-the-moment responses to complex real-world situations, which is a significant improvement to surveys confounded by retrospective recall. In addition, authors approach enables in-the-moment rating of signal pre-processing technology for real-world environments.

The results of this study have shown that program preference varied significantly among subjects. Also, it was noticed that to improve data collection, future research should encourage participants to realize a higher response rate. It was concluded that EMA is suitable for quantifying real-world self-reported preference, showing inter-subject variability in different listening environments.

The presented paper opens up new research capabilities in patients with cochlear implants.

  • We want to thank you for your effort.

Reviewer 3 Report

Thank you for your great work.

Although some demographic information was mentioned in the first paragraph of the materials and methods part,  it would be better to include demographic table in the manuscript result section. 

The manuscript is well written and easy to understand.  

 Unlike previous work that measured speech comprehension in a controlled sound room environment, the current research evaluated using ecological momentary assessment (EMA) to obtain in-the-moment responses from cochlear implant users. The new point which their results indicated is that EMA is suitable for quantifying real world self -assessment.  Previous research used EMA to assess acoustic environment and rating hearing experience. The novelty of this work is using EMA for speech understanding especially in fluctuating noise. Their conclusions were consistent with their findings although there were some limitations that should be considered in their future studies.

Author Response

Thank you for your great work.

Although some demographic information was mentioned in the first paragraph of the materials and methods part,  it would be better to include demographic table in the manuscript result section. 

  • We provided in table 1 an overview of demographic information. These data are age, usage of CI, side, gender, rate and maxima.

The manuscript is well written and easy to understand.  

 Unlike previous work that measured speech comprehension in a controlled sound room environment, the current research evaluated using ecological momentary assessment (EMA) to obtain in-the-moment responses from cochlear implant users. The new point which their results indicated is that EMA is suitable for quantifying real world self -assessment.  Previous research used EMA to assess acoustic environment and rating hearing experience. The novelty of this work is using EMA for speech understanding especially in fluctuating noise. Their conclusions were consistent with their findings although there were some limitations that should be considered in their future studies.

  • We want to thank you for your effort.